# Enhanced Biodegradation of Phenylurea Herbicides by *Ochrobactrum anthrophi* CD3 Assessment of Its Feasibility in Diuron-Contaminated Soils

**DOI:** 10.3390/ijerph19031365

**Published:** 2022-01-26

**Authors:** Lara-Moreno Alba, Morillo Esmeralda, Villaverde Jaime

**Affiliations:** Department of Agrochemistry, Environmental Microbiology and Soil Conservation, Institute of Natural Resources and Agrobiology of Seville, Spanish National Research Council (IRNAS-CSIC), 41012 Seville, Spain; lara@irnas.csic.es (L.-M.A.); morillo@irnas.csic.es (M.E.)

**Keywords:** phenylurea herbicides, 3,4-dichloroaniline, soil, bioremediation, bioaugmentation, biostimulation, 2-hydroxypropyl-β-cyclodextrin

## Abstract

The phenylurea herbicides are persistent in soil and water, making necessary the de-velopment of techniques for their removal from the environment. To identify new options in this regard, bacterial strains were isolated from a soil historically managed with pesticides. *Ochrobactrum anthropi* CD3 showed the ability to remove completely herbicides such as diuron, linuron, chlorotoluron and fluometuron from aqueous solution, and up to 89% of isoproturon. In the case of diuron and linuron, their main metabolite, 3,4-dichloroaniline (3,4-DCA), which has a higher toxicity than the parent compounds, was formed, but remained in solution without further degradation. *O. anthropi* CD3 was also tested for bioremediation of two different agricultural soils artificially contaminated with diuron, employing bioremediation techniques: (i) biostimulation, using a nutrient solution (NS), (ii) bioaugmentation, using *O. anthropi* CD3, and iii) bioavailability enhancement using 2-hydroxypropyl-β-cyclodextrin (HPBCD). When bioaugmentation and HPBCD were jointly applied, 50% of the diuron initially added to the soil was biodegraded in a range from 4.7 to 0.7 d. Also, 3,4-DCA was degraded in soil after the strain was inoculated. At the end of the soil biodegradation assay an ecotoxicity test confirmed that after inoculating *O. anthropi* CD3 the toxicity was drastically reduced.

## 1. Introduction

Over the few last decades, the use of pesticides in agriculture has become more extended. Studies have proved the presence of high levels of pesticides in soils, leachates and river water, higher than the limit values established by the local, national and European legislation [1]. Phenylurea herbicides are among the most widely pesticides used in agriculture worldwide [2]. They are used for pre- and post-emergence pest control in different crops such as cotton, cereals, and fruits [3]. These herbicides act as photosynthesis inhibitors by blocking electron transfer at photosystem II resulting in necrosis and plant death [4]. 

Their solubilities in water range generally from moderately to relatively high and they are expected to have low to moderate adsorptivity to soil, based on their octanol-water partition coefficient, K_ow_ [5] (Appendix A). This makes them mobile in soils and, once applied to the target field, irrigation and rainfall can deliver them through runoff and leaching from the ground to rivers, groundwaters, seawater and lakes [6]. However, the properties of phenylurea herbicides (Appendix A) significantly affect the possibility of their adsorption in soils, and, consequently, their mobility and their possibility of reaching water environments. Blondel et al. and Agbaogun et al. [7,8] observed that the K_ow_ values of these herbicides were as high as the number of chlorine atoms on the phenyl group, increasing their adsorption to soils. Consequently, diuron and linuron, with two chlorine atoms, present the highest adsorption to soils and are considered only as slightly mobile, and, therefore, they are the most frequently detected in contaminated soils. The rest of phenylurea herbicides are classified as mobile, being more frequently detected in water environments [9]. 

The degradation of phenylurea herbicides, both in water and in soil gives, as a result, the production of metabolites, such as N-(3,4-dichlorophenyl)urea (DCPU), N-(3,4-dichlorophenyl)-N-methylurea (DCPMU) and 3,4-dichloroaniline (3,4-DCA), which are more toxic than their parent contaminant, and together with them can lead to serious environmental and public health problems [10]. For these reasons, many phenylurea herbicides have been banned in European countries [11], where their presence in soils and water has to be progressively reduced, but they continue to used in numerous regions of the world. Therefore, the removal of these herbicides from the environment is a central issue. 

In the last decades, eco-friendly methods to remediate polluted environments have emerged. One of them is the use of microorganisms which can remove a wide variety of organic pollutants and can adapt to different bleak environments [12]. However, the natural degradation of these herbicides in soil is relatively slow. One of the reasons could be the insufficient number of endogenous degrading microorganisms specific for these herbicides, or that the endogenous microbiota does not have the catabolic capacities to metabolise and effectively degrade the target contaminants [13]. With the aim of improving degradation in soil, studies have been focused on isolation and identification of different specific degrading microbial populations from adapted soils which have showed capabilities to eliminate these organic contaminants from environments [13,14].

Diuron biodegradation in aqueous solution through microbial inoculation (bioaugmentation) has been studied by several authors, inoculating different microorganisms in the form of microbial consortia [15,16] or employing pure bacterial cultures, such as, *Arthrobacter* sp. N2 [17]; *Variovorax* sp. SRS16 [18] and *Micrococcus* sp. PS [19]. However, as far as we know, few studies have been described in literature about the use of bioaugmentation for diuron biodegradation in contaminated soils. Bacterial strains belonging to the genus *Bacillus* and *Stenotrophomonas* were employed in bioaugmentation [20,21]). In another work, the effect of the fungus *Neurospora intermedia* to degrade diuron in soil was studied by Villaverde et al. [22]. The remaining studies in soils were performed by Villaverde et al., directly using the endogenous soil microbiota as degrader [23], microbial consortia isolated from different soils [24] or an artificial consortium composed of *Arthrobacter sulfonivorans*, *Variovorax soli* and *Advenella* sp. JRO [25]. However, to the best of our knowledge, the bacterial strain *O. antropi* has never been described as a phenylurea degrader.

Although bioavailability of phenylurea herbicides is not considered as the main issue concerning biodegradation [26], the bioavailability in soils of most hydrophobic compounds is reduced with their ageing [27]. Cyclodextrins (CDs) have been suggested as an option for the removal of hydrophobic contaminants from soils because they are capable of increasing water solubility of hydrophobic organic compounds [28]. This ability to form inclusion complexes with different contaminant guest molecules has been employed in soil bioremediation processes [29] since CDs enhance the water solubility of low polarity compounds and hence, their bioavailability [30].

In this work, the strain *O. anthropi* CD3 was isolated in our laboratory, using diuron enrichment cultures, from an agricultural soil treated for decades with a variety of herbicides, including phenylureas. As previously mentioned, the species *O. anthropi* has never been described as a phenylurea degrader and degrades very high percentages of the five phenylureas studied, but, in addition, few studies have described bacterial strains belonging to this genus as pesticide degrader. *Ochrobactrum* sp. JAS2 was able to degrade 300 mg L^-1^ of chlorpyrifos after 12 h [31] or *Ochrobactrum intermedium* G3.48 was described as diuron-degrading (33% after 120 h) [32]. In the present study, CD3 strain was tested for diuron, linuron, chlorotoluron, isoproturon and fluometuron biodegradation in solution. Furthermore, the investigated strain was inoculated in two soils artificially contaminated with diuron, with the aim of finding an effective bioremediation tool based on biostimulation (using nutrients solution (NS)), bioaugmentation (inoculating with *O. anthropi* CD3) and the use of 2-hydroxypropyl-β-cyclodextrin (HPBCD) which is able to improve water solubility of the herbicide diuron, increasing it up to 23-fold [25], and hence, its bioavailability to be degraded. An ecotoxicology study was also carried out before and after soil diuron bioremediation by *O. anthropi* CD3 to determine if the treatment achieved the desired decrease in soil toxicity.

## 2. Materials and Methods

### 2.1. Materials

Analytical grade (99%) chlorotoluron (3-(3-chloro-4-methyl)-1,1-dimethylurea), isoproturon (3-(4-isopropylphenyl)-1,1-dimethylurea), fluometuron (1,1-dimethyl-3-[3-(trifluoromethyl)]phenylurea), linuron (3-(3,4-dichlorophenyl)-1-methoxy-1-methylurea), and the metabolite 3,4-dichloroaniline were purchased from Sigma-Aldrich (Madrid, Spain). Technical grade (98%) diuron (N-(3, 4-dichlorophenyl)-N, N-dimethylurea) was supplied by Presmar S.L. (Seville, Spain). HPBCD was provided by Cyclolab Ltd (Budapest, Hungary).

Three agricultural soil samples (LL, PLD, R) from the South of Spain were used, classified as an Alfisol, Inceptisol and Alfisol, respectively [33]. LL soil is located at Vejer de la Frontera—Cádiz (36°17′52.6″ N 5°52′45.2″ W), dedicated to the intensive agriculture of carrots, avocados, grapefruits, leeks, sunflowers, etc. This soil has been treated for many years with phenylurea herbicides and other organic pesticides such as organochlorine and organophosphate pesticides. PLD soil was taken from a farm located in Los Palacios y Villafranca—Seville (37°10′20.0″ N 5°55′21.9″ W) and has been treated with organohalogen herbicides for years, and wheat, cereals and vineyard has been grown on this soil. R soil was collected from a palm trees area in Conil de la Frontera—Cádiz (36°18′32.4″ N 6°08′58.6″ W) treated with of the insecticide chlorpyrifos. Soil samples were taken from the superficial horizon (0–20 cm) and were partially air-dried for 24 h and sieved (2 mm). Their properties are shown in Table 1. The pH was determined in a 1:2.5 soil/water extract. A Bouyoucos densimeter was used to evaluate the particle size distribution; OM was quantified by K_2_Cr_2_O_7_ oxidation; and the total carbonate content was measured using the manometric method. The water holding capacity of the soils was 52.4%, 23.16% and 53.2% for LL, PLD and R, respectively.

### 2.2. Diuron Degrader Enrichment

Diuron degrader bacterial strains were isolated in our laboratory from the agricultural soil sample LL. 10 g of LL soil was added to sterilized 250 mL Erlenmeyer flasks (JP-Selecta, Barcelona, Spain) with one cycle at 120 °C, inlet pressure of 101 kPa, for 20 min with 50 mL of a mineral salt medium (sterilized MSM), which provided the macronutrients. MSM contained (g L^−1^): 4.0 Na_2_HPO_4_; 2.0 KH_2_PO_4_; 0.8 MgSO_4_; 0.8 NH_4_SO_4_. The aqueous solution was spiked with diuron to reach a concentration of 1 g L^−1^, which was the only source of carbon and energy. 1 mL of a micronutrients solution (SNs), which contained: 12.5 NiCL_2_ 6H_2_O; 25.0 SnCl_2_ 2H_2_O; 12.5 ZnSO_4_ 7H_2_O; 12.5 Al_2_(SO_4_)_3_ 18H_2_O; 75.0 MnCl_2_ 4H_2_O; 12.5 CoCl_2_ 2H_2_O; 37.5 FeSO_4_ 7H_2_O; 10.0 CaSO_4_ 2H_2_O; 3.75 KBr; 3.75 KCl; 2.50 LiCl (mg L^−1^) was also added [34]. The mixture of SNs and MSM (1:50) was named nutrient solution (NS). Incubation conditions were 150 rpm at 30 °C, and every 7 d (four times) an aliquot of 10 mL of the culture was removed and transferred to another flask containing the same previously described composition. Aliquots of 100 µL of the final culture were added on agar plates, prepared with MSM medium and diuron with a concentration of 0.5 g L^−1^. Successive isolations were performed, and different colonies, according to their size, colour, edge and elevation, were selected. Eleven strains (LL1D, LL2D, LL3D, LL4D, LL5D, LL6D, LL7D, LL8D, LL9D, LL10D and LL11D) were isolated. The selected isolated strains were kept in cryovials (Microbank^®^, Pro-Lab Diagnostics, Bromborough, UK), which contained a specific culture medium and 20 porous spheres at 3 mm diameter. They were kept at −80 °C.

### 2.3. Inoculum Preparation 

For each experiment, bacterial cryovials were defrosted and the bacteria were sowed in Luria−Bertani (LB) medium. The culture was incubated at 30 °C in an orbital shaker (150 rpm). The different bacterial strains were collected at the end of their respective exponential phase and later washed twice with a sterile MSM solution before beginning the experiments. The final density of the strains added in biodegradation assays was 10^8^ CFU mL^−1^.

### 2.4. Diuron Degrading Capacity of the Isolated Bacterial Strains 

The capacity of the 11 isolated strains to degrade diuron in solution was tested. Each inoculum was added to NS solution containing diuron as the only carbon and nitrogen sources (10 mg L^−1^) The samples were incubated in a thermostatic chamber at 30 °C for 20 d, and periodically the concentration of diuron was measured by HPLC as described in the Analytical Methods section.

### 2.5. Isolated Bacterial Strain Identification with 16S rDNA Amplification

The isolated strain which showed to be the most effective for diuron degradation in solution (Appendix A) was identified. DNA was extracted from the liquid cultured of the strain employing the G-spinTM total DNA Extraction Kit (iNtRON Biotechnology, Seongnam, Korea), then 16S rRNA gene was amplified by polymerase chain reaction (PCR) using a high-fidelity polymerase (Velocity DNA polymerase from Bioline, Almería, Spain) with universal oligonucleotides primers: 16F27 (annealing at position 8-27 *E. coli* numbering) and 16R1488 [35]. Finally, the PCR product was purified using a NucleoSpin® PCR clean-up gel extraction kit and PCR clean-up (Macherey-Nagel, Cultek S. L., Madrid, Spain) prior to be sequenced.

### 2.6. Biodegradation Experiments in Solution

Phenylurea herbicide biodegradation experiments in solution were performed in 25 mL sterilized glass vials during 60 d of assay. At each incubation timepoint (1, 3, 7, 12, 21, 30 and 60 d), aliquots were taken from solutions in triplicate. Each vial contained 330 µL of the bacterial inoculum (final density of 4.5 × 10^6^ CFU mL^−1^), 15 µL SNs and 15 mL of MSM, contaminated with 10 mg L^−1^ of each phenylurea herbicide. To monitor abiotic degradation, non-inoculated vials (control) were also set up. The vials were kept at a temperature of 30 °C in a laboratory incubator. Due to the importance and toxicity reported in the literature the main metabolite of diuron and linuron, 3,4-DCA, was analyzed at different time points. The samples were taken under sterile conditions in a vertical laminar flow cabin and filtered using a nylon filter (0.45 µm, RephiQuik, Madrid, Spain) for analytical analysis (see Section 2.10).

### 2.7. Biodegradation Experiments in Soils

Diuron biodegradation experiments in soils were performed in a 250 mL sterilized flask containing 100 g of soil sample contaminated with 50 mg kg^−1^ diuron and the required volume of NS (MSM + SNs) to reach 40% of the soil water holding capacity [24]. The treatments were assigned completely at random so that each experimental unit had the same chance of receiving any one treatment. Different bioremediation strategies were designed: (i) biostimulation: spiked soil + NS; (ii) bioaugmentation: spiked soil + NS + O. anthropi CD3inoculum (1 × 10^8^ CFU g^−1^); (iii) bioavailability enhancing: spiked soil + HPBCD solution + NS. HPBCD was added with a concentration corresponding to 10 times that of the diuron initially spiked in the soil sample; (iv) Combined use of biostimulation, bioaugmentation and bioavailability enhancing: spiked soil + HPBCD + *O. anthropi* CD3 + NS. In parallel, abiotic controls were set up adding 200 mg L^−1^ of HgCl_2_ to eliminate the soil endogenous microbiota in order to observe any diuron dissipation due to abiotic processes. All experiments were set at a temperature of 30 °C in a laboratory incubator for 100 d. Soil samples were taken at different times of the biodegradation assays (0, 6, 8, 13, 20, 27, 36, 47, 65, 87 and 100 d). Diuron degradation was measured through the relative decline of its residues in the soil samples. For this purpose, 1 g of each soil sample was exhaustively extracted with 5 mL of methanol. The extraction process was carried out through the following steps; (1) 24 h of agitation of the tubes in an orbital shaker at 100 rpm at 20 ± 1 °C, (2) 1 min of vortex mixer, (3) 30 min in an ultrasound bath and finally the supernatant was recovered by 10 min centrifugation at 8000 rpm. Once the samples were obtained, diuron and its metabolite 3,4-DCA were analysed by liquid chromatography as described in the Section 2.10.

### 2.8. Model of Biodegradation Kinetics

The biodegradation curves were fitted to a simple first-order kinetic model (SFO) and a biphasic first-order kinetic sequential model (Hockey- stick, HS) using an excel file provided by focus [36] and the Solver tool (Microsoft Statistical Package, New York, NY, USA), and the following equations:[C]_t_ = [C]_0_ e^−kt^ (SFO)
DT_50_ = ln 2/k (SFO)
[C]_t_ = [C]_0_ e^−k1t^b e^−k2(t-tb)^ (HS)
DT_50_ = (ln 100/100-50)/k_1_ if DT_50_ ≤ tb (HS)
DT_50_ = tb + (ln (100/100-50) − k_1_tb)/k_2_ if DT_50_ > tb (HS)
where [C]_t_ and [C]_0_ are the concentrations of diuron at time t and just at the beginning of the experiments. In SFO model, K is the degradation (d^−1^). DT_50_ is the time required for the pollutant concentration to decline to half of its initial value. HS model has two different rate constants of degradation, k_1_ and k_2_, and tb, which is the time at which the rate constant changes. SFO and HS models were chosen for their simplicity and their capabilities to fit previously published measured dissipation kinetic datasets for herbicides [37]. The coefficient of determination and the chi-square (χ^2^) test were calculated as indicators of the goodness of fit χ^2^ values less than 15 imply a good fit) [36].

### 2.9. Diuron Availability in Soil

Soil diuron extraction assays were carried out to check the effect of using NS (MSM + SNs) and HPBCD as extractants on diuron availability. One g of soil previously contaminated with 50 mg kg^−1^ of diuron was extracted with 5 mL of NS or NS combined with HPBCD (10 times the concentration of diuron initially spiked in soil). The centrifuge tubes were shaken on an orbital shaker for 72 h at 20 ± 1 °C, centrifuged (10 min, 7000 rpm) to separate the supernatant, which was filtered through a 0.22 μm glass fiber membrane (Millipore, Burlington, MA, USA). Diuron concentration was measured using the analytical methods described below.

### 2.10. Analytical Methods

Concentrations of the phenylurea herbicides and the metabolite 3,4-DCA during biodegradation experiments in soil and solution were analyzed on a LC-2010A HT HPLC system equipped with a UV detector (Shimadzu Scientific Instruments, Columbia, MD, USA). Kromasil C18 reverse-phase was the chromatographic column used, the mobile phase was acetonitrile:water (60:40, *v*/*v*), and all the compounds were detected by UV absorbance at 230 nm. The calibration curves were linear over the concentration range studied, with correlation coefficients higher than 0.99, indicating good performance of the chromatographic method. The detection limits calculated using a noise-signal ratio of 3 were below 0.01 mg L^−1^ for the investigated compounds, and the standard deviations were around 5%. In soils, diuron extraction with methanol was effective with recovery rates from 88 ± 6 to 115 ± 8% and relative standard deviation (RSD) lower than 15%. For the acquisition and management of the data, the LC Shimadzu Solution Chromatography Data System computer program was used.

### 2.11. Toxicity Analysis

The Microtox^®^ Test System (Modern Water Inc., New Castle, U.K.) based on the bioluminescent of the bacterium *Vibrio fischeri* was used to measure the toxicity of the soil, according to the standard protocol for the Microtox^®^ basic test [38]. This toxicity test can be applied to liquid waste discharged in natural waters and also to leachates/percolates from contaminated soils. Soil toxicity was estimated through the determination of the calculated value EC_50_ of the soil lixiviate, which is a hypothetical value that represents the soil extract concentration (% *v*/*v*) having a toxic effect that would produce 50% reduction in luminescence in *V. fischeri*. Briefly, 2 g of each soil sample was added to 3 mL of NaCl at 2% solution. These suspensions were hatched in an orbital shaker for 10 min, centrifuged for 2 min at 10,000 rpm and serially diluted (1:2) with NaCl at 2% solution (6.25%, 12.5%, 25% and 50% (*v*/*v*)) and compared with the control [39]. Freeze-dried *V. fischeri* were rehydrated forthwith prior to use in testing. Tests were performed in a temperature-controlled photometer at 15 °C (Microbics Inc., Lenexa, KS, USA, 1992). It was assumed that the control containing only *V. fischeri* is considered to have 0% of inhibition, and the lixiviate containing only NaCl (2%) is considered to have 100% of inhibition. EC_50_ parameter is given by the Microtox® Text System for each sample analysed, and the toxic units (TU) were estimated using the formula TU = 100/EC_50_. TU results were classified according to Persoone et al. [40] (TU < 0.4 no acute toxicity; 0.4 < TU < 1 light acute toxicity; 1 < TU < 10 acute toxicity; 10 < TU < 100 high acute toxicity; TU > 100 very high acute toxicity). 

### 2.12. Statistical Analysis

One-way ANOVA with three replicates was performed to study the significant differences among treatments. Statistical analysis was conducted by means of the SPSS v. 21, (SPSS Inc., Chicago, IL, USA) statistical package.

## 3. Results and Discussion 

### 3.1. Identification of the Diuron Degrading Bacterial Strain 

A soil that had been treated with a great variety of pesticides for a long time, including phenylurea herbicides, was enriched with diuron in order to isolate a diuron-degrading mixed consortium. 11 bacterial strains (LL1D, LL2D, LL3D, LL4D, LL5D, LL6D, LL7D, LL8D, LL9D, LL10D and LL11D) were isolated from the consortium initially obtained and only the strain LL6D showed ability to remove diuron in solution, achieving an extent of biodegradation 97.4% after 20 d of the assay (Appendix A). LL6D was identified and selected to perform biodegradation assays due to its capacity to use diuron as the only source of carbon and energy. 16S rRNA gene sequence showed a match of 100% to species from *O. anthropi* in the NCBI GenkBank, and it was named *O. anthropi* CD3. This strain belongs to the phylum Proteobacteria. Other bacteria belonging to this phylum are in relation with phenylurea herbicides biodegradation. Pesce et al. [41] showed that Proteobacterium phylum was identified as the bacterial phylotype after being in contact with a realistic diuron exposure. *Sphingomonas* sp. strain SRS2, which belongs to the Proteobacterium phylum, combined with SRS1 strain isolated from a soil contaminated by isoproturon, were able to mineralize the herbicide diuron. SRS2 was described as a possible new genus whose 16S rRNA gene sequence present 95.7% of similarity with *Polaromonas vacuolata* [42]. However, it is important to highlight that *O. anthropi* has never been described as a phenylurea degrader, although several studies have described strains belonging to *Ochrobactrum* species as degrader for other pesticides. *Ochrobactrum* sp. JAS2 was used to degrade chlorpyrifos, an organophosphate insecticide, in an aqueous medium [31]. *O. anthropi* strain NC-1, isolated from a pesticide-contaminated soil, was identified as a degrading strain (98.5% within 7 d) for phenmedipham [43]. *O. thiophenivorans* and *Sphingomonas melonis* were inoculated forming a consortium to remove the insecticide methomyl, achieving a bioremediation rate of 86% [44].

### 3.2. Phenylureas Biodegradation in Solution by O. anthropi CD3

*O. anthropi* CD3 isolated from an agricultural soil after enrichment with diuron, has been tested for biodegradation of five herbicides belonging to the phenylurea family. *O. anthropi* CD3 biodegraded, to different extents, all the studied herbicides (Figure 1). All the obtained biodegradation curves were fitted to a simple first-order kinetic model (SFO) (Table 2). In the case of isoproturon, a significant percentage of biotransformation was observed reaching a degradation of about 89% after the 60 d of the assay (Table 2). In the cases of diuron, linuron, chlorotoluron and fluometuron, their initial concentration was completely removed when *O. anthropi* CD3 was inoculated. Moreover, diuron, linuron and chlorotoluron were rapidly transformed, achieving 50% of biodegradation after only 1.1, 3.8 and 0.4 d, respectively (Table 2). 

The relationship between pesticide structure and biodegradation is known. Alexander [45] showed that there is the specificity of the microorganism for a source of carbon is mainly related to the enzymatic function, which is largely restricted to performing only a single type of reaction on a range of substrates of very similar structures. In this work, *O. anthropi* CD3 was able of degrading five herbicides that belong to the same family and, therefore, with similar chemical structures. 

The ability of individual strains to degrade different phenylurea herbicides have been previously demonstrated. Tixier et al. [17] carried out biodegradation experiments in solution employing the bacterial phenylureadegrader *Arthrobacter* sp. N2, isolated from a diuron-contaminated soil. The ability of *Arthrobacter* sp. N2 to degrade diuron, chlorotoluron and isoproturon in solution depended on the herbicide considered. Although the three phenylurea herbicides were completely degraded, the fastest biodegradation occurred with diuron (10 h), followed by chlorotoluron (30 h) and finally isoproturon (5 d). In the case of this last herbicide, the transformation rate seemed to be affected by the presence of the isopropyl group. Cullington and Walker [46] also studied this phenomenon during biodegradation assays of five different herbicides in solution using an unidentified soil bacterium, observing different efficacy in the rate of degradation: linuron > diuron > monolinuron >> metoxuron >>> isoproturon. Diuron and linuron were easily converted whereas the bacterium showed much lower activity towards isoproturon. Something similar happened in the present study, where isoproturon was the only phenylurea herbicide that was not completely degraded after 60 d of incubation, which pointed out that the presence of the isopropyl group influenced the transformation rate. The isopropyl group makes it difficult for microorganisms to use isoproturon as a carbon source, because it presents steric hindrance and hence microorganisms will prefer other easier to attack carbon sources.

Sharma et al. [19] reported that *Micrococcus* sp. PS-1 isolated from a diuron storage site could biodegrade diuron and other structural analogue phenylurea herbicides in the order: fenuron (99%) > monuron (98%) > diuron (96%) > monolinuron (80%) > linuron (74%) > chlorotoluron (65%). These results suggested that the size, degree of ring chlorination and the nature of the urea N’-substituents were the determining factors in the occurrence of degradation of phenylurea herbicides. Similarly, in our study, different biodegradation rates were observed, probably due to their structural differences.

The structure of phenylurea herbicides is not the only factor influencing their biodegradability as mentioned above, but also the type of bacterial strain plays an important role. The Gram-negative strain *Diaphorobacter* sp. LR2014-1 was capable of biodegrading five phenylurea herbicides with different structures [16]. The authors concluded that this large substrate spectrum points to the fact that strain LR2014-1 harbours an unusual enzyme or multiple enzymes to catabolize these phenylureas. It should be pointed out that those herbicides which presented the highest degradation (linuron and chlorotoluron) were those which gave the lowest degradation in the study of Sharma et al. [19], indicating the important role of the catabolizing enzymes present in the bacterial strain used. 

As with the biodegradation observed with the bacterial strains reported previously, the results obtained indicate that the isolated bacterial strain *O. anthropi* CD3 has an exceptionally broad substrate spectrum for a variety of substituted phenylurea herbicides. 

In this work, 3,4-DCA was the only metabolite quantifiable in the solution formed from diuron and linuron biodegradation after inoculation with the investigated phenylurea degrading strain. For chlorotoluron, fluometuron and isoproturon typical aniline metabolites (3-chloro-4-methylaniline, *m*-trifluoromethylaniline, isopropylaniline, respectively) were not detectable at the end of the biodegradation assay. Also, Tixier et al. [17] observed that the aniline metabolites from chlorotoluron and isoproturon were produced at a low concentration that they could not be also detected, whereas formation of 3,4-DCA from diuron was almost quantitative. 3,4-DCA can be produced during biotic and abiotic degradation processes of diuron and linuron, and shows a higher ecotoxicological impact on living organisms than the parent compounds and hence the importance of studying its possible formation. Huda Bhuiyan et al. [47] stated that 3,4-DCA can be considered as one of the major aniline derivatives released into the environment, capable of provoking disruption of the endocrine and other biological systems in animals. Figure 2 shows the concentration of 3,4-DCA produced throughout the biodegradation of diuron and linuron in solution when *O. anthopi* CD3 was inoculated. The highest 3,4-DCA concentration was observed within 9 and 15 d of incubation in the case of diuron (4.7-4.9 mg L^−1^), and in the case of linuron, the highest concentration (5.9 mg L^−1^) was reached after 7 d of assay. 3,4-DCA concentration was maintained without any further degradation until the end of the experiment (60 d). In both cases a similar metabolite formation profile was observed.

### 3.3. Diuron and 3,4-Dichloroaniline Biodegradation in Soils

*O. anthropi* CD3 has been also selected for performing diuron biodegradation experiments in soils. As commented above, Blondel et al. and Agbaogun, et al. [7,8] observed that the K_ow_ values of phenylurea herbicides were as high as the number of chlorine atoms on the phenyl group, provoking an increase in their adsorption to soil particles. This property combined with the high level of persistence of the herbicide diuron makes it a hazard for soil environment. 

Different bioremediation techniques have been proven for application in two different soils (PLD and R), which were artificially contaminated with diuron: (i) biostimulation, using a nutrient solution (NS) which accelerates the decontamination rate; (ii) use of HPBCD solution as bioavailability enhancer; (iii) bioaugmentation, using *O. anthropi* CD3. Figure 3 shows the different biodegradation curves obtained from these assays. 

Diuron abiotic dissipation was evaluated in both soils, where a HgCl_2_ solution was added to soil prior to begin of the assays with the aim of killing the soil endogenous microbiota (Figure 3). The global extent of biodegradation for the abiotic control was only 1.1% and 1.2% for R and PLD, respectively (Table 3). The role of soil microbiota in diuron biodegradation was evaluated through the addition of nutrients (NS), but no significant biodegradation was observed in any of the investigated soils (1.3% after 100 d), with DT_50_ values of 24.6 and 29 years for soils PLD and R, respectively, indicating the extremely high persistence of diuron in these soils. This observed result could be due to the fact that the soil endogenous microbiota is not adapted for diuron degradation; but another reason can be possible: the low bioavailability of the herbicide due to its strong adsorption on soil particles. These results point the need to use techniques of bioaugmentation with specific microbial degraders and/or the use of CDs as bioavailability enhancers. 

This latter option was tested by using HPBCD. In the case of R soil, diuron biodegradation curve obtained in the presence of NS and HPBCD indicated a slight enhancement in comparison with the biodegradation profile obtained when only biostimulation with NS was applied (Figure 3A). DT_50_ decreased from more than 10,000 d until about 1000 d, and 20.3% of diuron biodegraded was reached after 100 d of experiment (Table 3). In the case of PLD soil, better results were observed, where the addition of HPBCD provoked a great increase in diuron biodegradation, decreasing DT_50_ from almost 9000 d to about 37 d, and reaching an extent of biodegradation of 84%. These results might be due to the low OM content in PLD soil (1.67%) in comparison with the R soil (3.44%). Diuron has a high affinity for soil OM, causing a reduction of its bioavailability as higher is the percentage of OM in soil [48,49,50].

HPBCD is widely known as an ecofriendly compound for organic contaminants bioremediation due to its ability to form inclusion complexes between HPBCD hydrophobic cavity and hydrophobic molecules, which improves their water solubility, making them more available for microorganisms [51,52,53]. In this case, the increase in soil availability of diuron mediated by HPBCD will be facilitated as lower is the OM content of the soil.

Reid et al. [54] conducted a non-exhaustive extraction technique employing HPBCD solutions for PAHs, and observed a significant correlation with mineralization of a phenanthrene-degrading bacteria in three different soils to assess its bioavailability. This method has been broadly used for assessing the availability of different types of organic contaminants in soil [23,55]. 

Soil diuron extraction experiments were performed, in order to test the effect of soil solution, NS and HPBCD solutions on diuron availability on the two investigated soils (Figure 4). The percentage of diuron extracted was lower when NS solution was used as extractant. When HPBCD was employed, in the case of the R soil, the amount of diuron extracted was similar to those extracted with the NS solution 17.4% and 22.3%, respectively, and in the case of PLD soil with a lower soil OM content, diuron was extracted from 25.4%, when NS was added, to 58% when HPBCD was used. The content of OM in the soil plays an important role in diuron bioavailability [27], which explains the significant decrease observed in DT_50_ for diuron biodegradation in the case of PLD soil (Table 3).

The higher diuron degradation observed can also be caused by the stimulating effect of HPBCD on the soil microbial community, since it is an oligosaccharide, and may be considered an attractive carbon source for microorganisms [56]. The high percentage of degradation obtained in PLD soil after the addition of HPBCD indicates that the soil microbiota would act as an effective diuron degrader after being stimulated in the presence of HPBCD, because such microbiota had been previously adapted to the presence of diuron or other compounds with similar chemical structures, which would have produced an alteration of soil microbial communities, favoring the development of some microbial taxa which are able to use diuron as a source of carbon and energy [57].

The effect of bioaugmentation with *O. anthropi* CD3 on diuron biodegradation in both investigated soils can be also observed in Figure 3 and Table 3. In the case of R soil, the initial concentration of diuron was biodegraded completely after 50 d in the presence of *O. anthropi* CD3 + NS, and DT_50_ was 6.1 d. When *O. anthropi* CD3 and HPBCD were jointly applied, 100% of diuron was removed after 35 d of the assay, with a slight decrease in the DT_50_ (4.7 d). In the case of PLD soil, diuron was completely biodegraded when *O. anthropi* CD3 + NS was applied after 20 d of the assay (Figure 3), and the DT_50_ observed was only 3.1 d. A particularly noteworthy improvement in biodegradation kinetics was observed when the bacterial strain was added together with HPBCD, decreasing DT_50_ to only 0.7 d. 

The main metabolite of diuron, 3,4-DCA, was monitored throughout the soil biodegradation assays when using *O. anthropi* in both soils (Figure 5 and Appendix A). Its maximum concentration in PLD soil was observed after 24 h of the assay (Figure 5B), reaching a higher value in the presence of HPBCD, and being completely degraded after 50 d. R soil has a higher soil OM content than PLD soil, which provoked a lower bioavailability of both, the herbicide and 3,4-DCA [58], and for this reason, the maximum concentration of 3,4-DCA was reached after 8 d, and its dissipation rate in this soil was much slower than in PLD soil, being not possible to achieve its complete biodegradation after 100 d of incubation (Figure 5). 

*O. anthropi* CD3 has proved to be capable of removing completely the initial concentration of diuron in both studied soils. However, this bacterial strain was not capable of carrying out the degradation of 3,4-DCA in solution (Figure 2). The synergetic action of *O. anthropi* CD3 and the soil endogenous microbiota was able to reduce the metabolite concentration formed during diuron biodegradation assays. The microbial community of soil PLD showed to be more effective to degrade diuron and its metabolite than that of R soil, due to the contact of PLD soil with compounds with chemical structures similar to diuron for a longer time, as commented above. 

The significance of bacterial consortia composed of several strains with synergistic catabolic co-operation for the degradation of multiple pesticides in solution has been previously mentioned [59].

### 3.4. Ecotoxicology

Soil toxicity was evaluated in the studied artificially contaminated soils (50 mg kg^−1^ of diuron) before and after treatment with *O. anthropi* CD3 and/or HPBCD solution using a Microtox^®^ test. The measured toxicity parameters are shown in Table 4. In the case of the R soil, after inoculation and inoculation + HPBCD only a slight decrease in toxicity at the end of the experiment (100 d) was determined. TU value before inoculation was 4.0, and this toxicity was reduced to 3.2 and 3.3, respectively, after both treatments were applied. Even though diuron was completely biodegraded, a level of acute toxicity was maintained as a result of the presence of the formed 3,4-DCA remaining in the soil after being treated (Figure 5A). In R soil, its endogenous microbiota was not able to synergistically act with the investigated strain *O. anthropi* CD3 to completely remove the metabolite. On the contrary, in the case of PLD soil, significant differences could be observed in the values studied as indicators of toxicity. When *O. anthropi* CD3 was inoculated, TU decreased from 10.9 (high acute toxicity, 10 < TU < 100) to 1.4 (acute toxicity), a value very close to slight acute toxicity (0.4 < TU < 1). Similar results were observed when HPBCD and *O. anthropi* CD3 were employed together for soil decontamination. This strong toxicity decrease is due to the fact that 3,4-DCA was completely removed from PLD soil sample (Figure 5B), but, in spite of this, the soil remained toxic, indicating that other toxic degradation compounds must be present, and, therefore, the soil sample would need a deeper bioremediation to decrease its toxicity to levels similar to the uncontaminated soil. Monitoring pollutant levels by analytical methods shows the degradation of the contaminants, but it does not reveal the toxicity of the potentially formed metabolites. For this reason, it is important to assess the soil ecotoxicology after any bioremediation technique application [60,61] to determine whether or not there is toxicity in the sample, through their comparison with the classification proposed by Persoone et al. [40].

## 4. Conclusions 

The Gram-negative bacterial strain *O. anthropi* CD3 isolated from an agricultural soil by selective enrichment with diuron, and which was not previously described as phenylurea degrader, was capable of utilizing some phenylurea herbicides as sole carbon sources. The relative degradation profile in aqueous solution of *O. anthropi* CD3 was in the order of chlorotoluron > diuron > linuron > fluomethuron > isoproturon, which was related to the different structural features of the molecules. The only metabolite detected throughout the biodegradation processes was 3,4-DCA from diuron and linuron, which could not be degraded by *O. anthropi* CD3 in solution. 

The use of *O. anthropi* CD3 in soils contaminated by diuron led to its degradation, but at different levels depending on their soil organic matter content (the higher the OM the lower the biodegradation) and the characteristics of their endogenous microbiota, reaching the complete degradation of diuron and 3,4-DCA in that soil with a microbial population more adapted to degrade similar contaminants and able to synergistically act with *O. anthropi* CD3 to remove the metabolite. HPBCD was used as a bioavailabity enhancer of diuron due to its capacity to form an inclusion compound with the herbicide, and also as biostimulant of the endogenous microbial community, increasing considerably the rate of diuron biodegradation in the soil with the least amount of OM.

Ecotoxicity studies showed a decrease in toxicity of the studied contaminated soils, but more drastically in the soil where both diuron and 3,4-DCA were completely removed. However, the complete elimination of toxicity could not be reached, indicating the presence of other toxic compounds. It indicates the need to carry out ecotoxicity studies to assess the feasibility of any bioremediation treatment application, even when the parent contaminant and its main metabolites had been reduced or completely eliminated.

## Figures and Tables

**Figure 1 ijerph-19-01365-f001:**
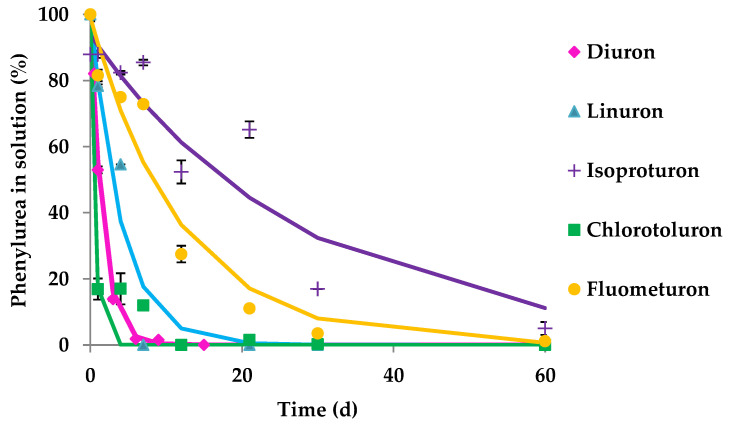
Biodegradation curves (60 d) of (♦) diuron, (▲) linuron, (+) isoproturon, (■) chlorotoluron and (●) fluometuron in solution in presence of *Ochrobactrum anthropi* CD3. Solid lines show model fitting to the experimental results (symbols). Standard deviation (vertical bars).

**Figure 2 ijerph-19-01365-f002:**
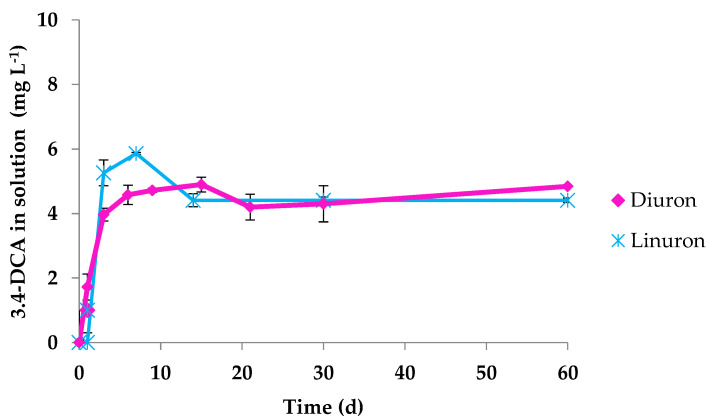
Concentration of 3,4-dichloroaniline (3,4-DCA) in solution throughout the biodegradation process of diuron (♦) and linuron (*****) in presence of *Ochrobactrum anthropi* CD3. Standard deviation (vertical bars).

**Figure 3 ijerph-19-01365-f003:**
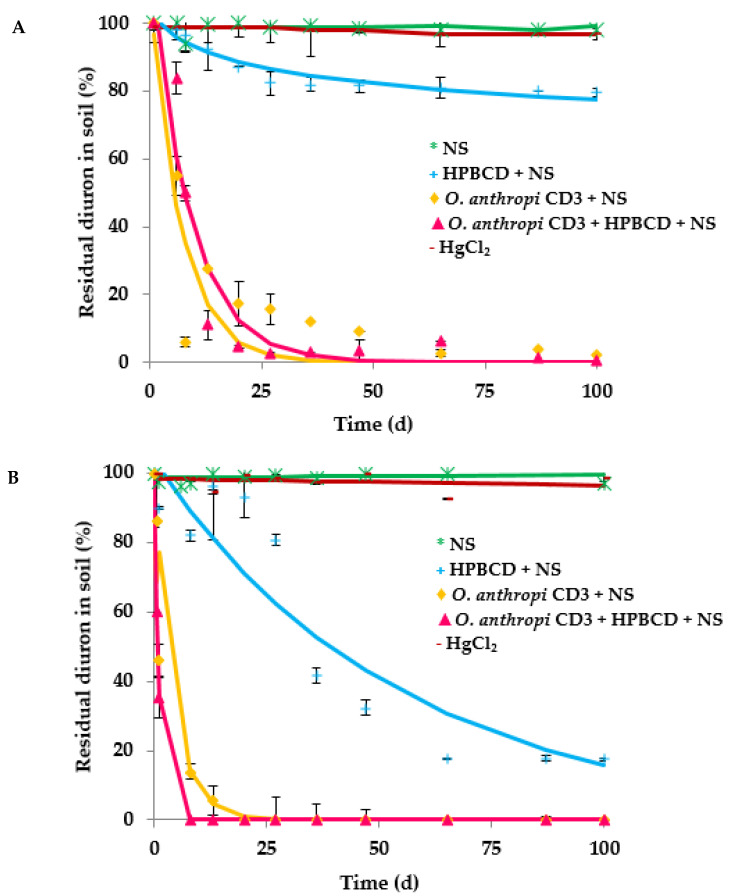
Diuron biodegradation curves in R (**A**) and PLD (**B**) soils after the application of: NS (*), HPBCD + NS (+), *Ochrobactrum anthropi* CD3 + NS (♦), *Ochrobactrum anthropi* CD3 + HPBCD + NS (▲) and HgCl_2_ (**-**). Standard deviation (vertical bars).

**Figure 4 ijerph-19-01365-f004:**
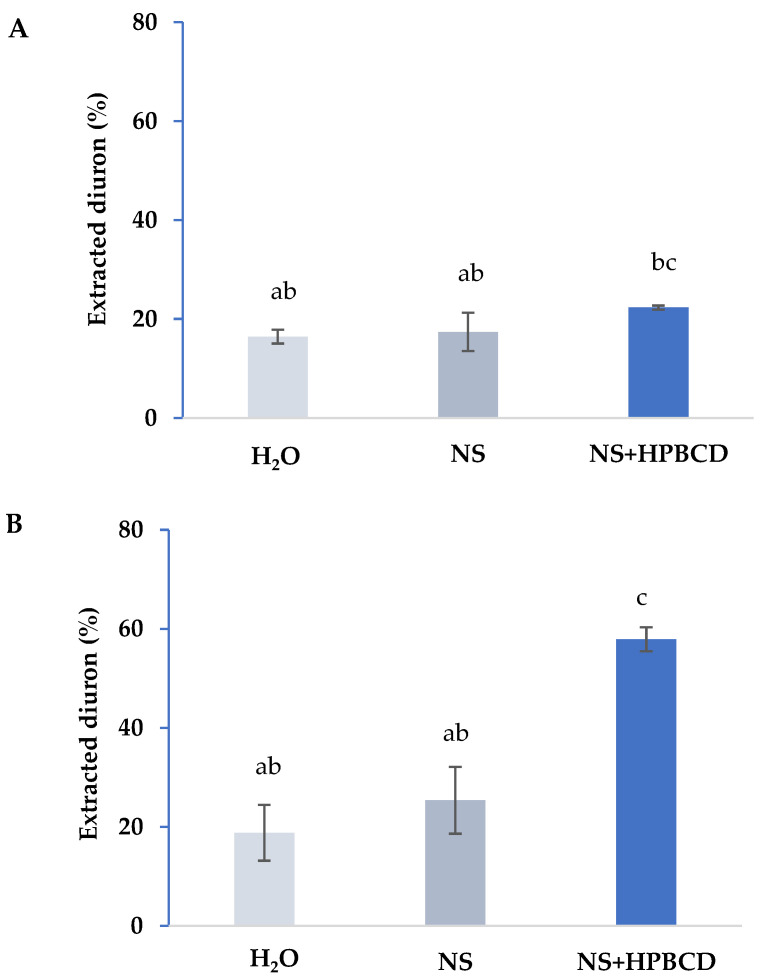
Diuron extracted from R (**A**) and PLD (**B**) soil in presence of H_2_0, nutrient solution (NS) and HPBCD. Standard deviation (vertical bars). The same letter in the same column means no significant differences among treatments.

**Figure 5 ijerph-19-01365-f005:**
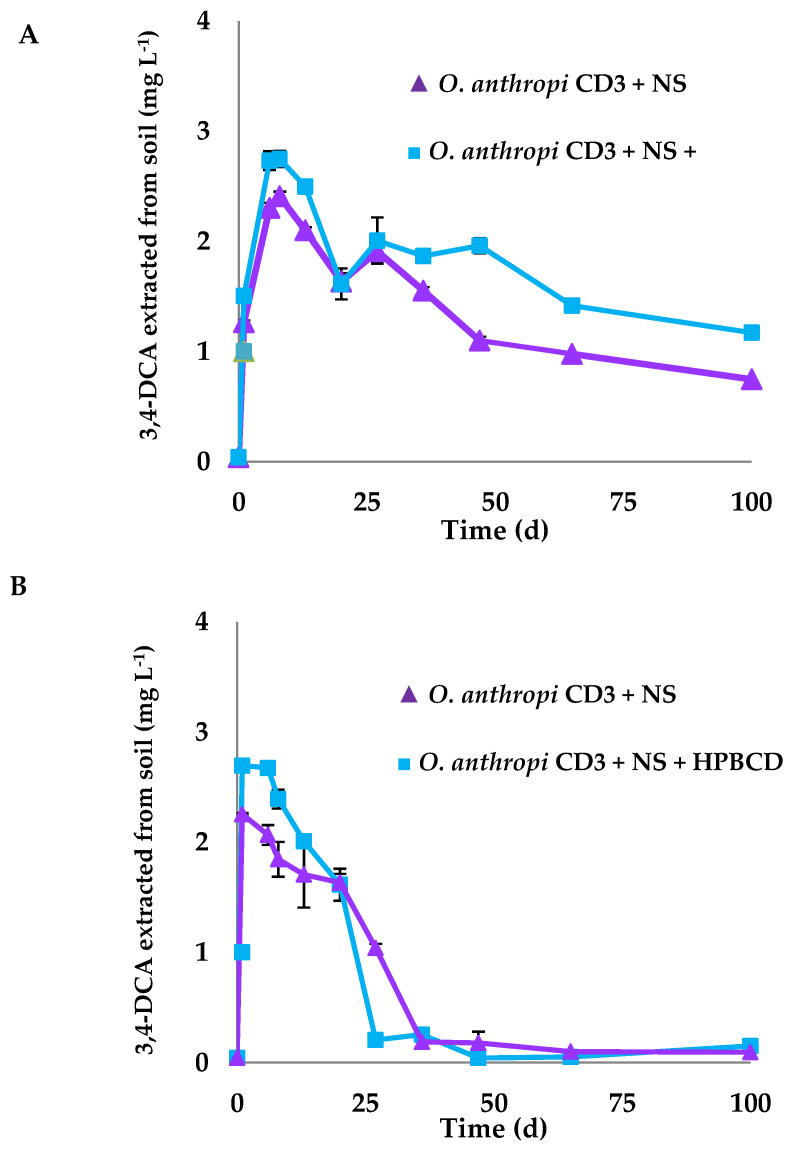
Concentration of 3,4-dichloroaniline (3,4-DCA) in soils R (**A**) and PLD (**B**) throughout the biodegradation process of diuron in presence of *Ochrobactrum anthropi* CD3 + NS (▲) and *Ochrobactrum anthropi* CD3 + HPBCD + NS (**■**). Standard deviation (vertical bars).

**Table 1 ijerph-19-01365-t001:** Some properties of the soils used.

Soils	pH	CO_3_^−2^ (%)	OM (%)	Sand (%)	Silt (%)	Clay (%)	Textural Classification	Taxonomic Classification *
R	7.72	4.00	3.44	77	9.50	13.5	Sandy	Alfisol
PLD	8.24	9.70	1.67	47.0	18.5	34.5	Clay loam	Inceptisol
LL	7.84	4.00	0.87	79.6	9.30	11.1	Sandy	Alfisol

* USDA SOIL TAXONOMY, Soil maps, 2005. National Geographic Institute. Nature Database (Spanish Ministry of Environment).

**Table 2 ijerph-19-01365-t002:** Kinetic parameters obtained from phenylurea herbicides biodegradation in solution after inoculation with *Ochrobactrum anthropi* CD3.

Contaminant	R^2^	χ^2^	K (d^−1^)	DT_50_ (d)	Extent of Biodegradation (%)
Diuron	0.89	7.2	6.2 × 10^−1^	1.1	100
Linuron	0.91	10.8	2.5 × 10^−1^	3.8	100
Isoproturon	0.94	14.6	3.5 × 10^−2^	19.5	89
Chlorotoluron	0.99	13.5	1.8	0.4	100
Fluometuron	0.98	14.3	8.4 × 10^−2^	8.3	100

Biodegradation curve were fitted to single first order model. K: mineralization rate constants. DT_50_: Time required for the concentration to decline to half of the initial value.

**Table 3 ijerph-19-01365-t003:** Kinetic parameters obtained after different biodegradation treatments applied in R and PLD soils contaminated with diuron.

Soil	Treatment	Kinetic Model	R^2^	χ^2^	K_1_ (d^−1^)	K_2_ (d^-1^)	tb (d)	DT_50_ (d)	Extent of Biodegradation (%)
	NS	SFO	0.99	4.2	5.2 × 10^−4^	-	-	10,569	1.3
	HPBCD + NS	HS	0.91	3.8	8.4 × 10^−3^	4.4 × 10^−4^	26.9	1077	20.3
R	*O. anthropi* CD3 + NS	SFO	0.99	2.3	1.1 × 10^−1^	-	-	6.1	100
	*O. anthropi* CD3 + HPBCD + NS	SFO	0.94	14.1	1.5 × 10^−1^	-	-	4.7	100
	HgCl_2_	SFO	0.98	5.1	5.1 × 10^−4^	-	-	17,011	1.1
	NS	SFO	0.99	7.9	4.2 × 10^−4^	-	-	8977	1.3
	HPBCD + NS	SFO	0.88	14.8	1.8 × 10^−2^	-	-	37.1	84.0
PLD	*O. anthropi* CD3 + NS	SFO	0.9	14.3	2.2 × 10^−1^	-	-	3.1	100
	*O. anthropi* CD3 + HPBCD + NS	SFO	0.92	14.8	1.1	-	-	0.7	100
	HgCl_2_	SFO	0.98	1.7	1.9 × 10^−4^	-	-	3501	1.2

NS: nutrients solution. K: mineralization rate constants. tb: Time at which rate constant changes. DT_50_: Time required for the concentration to decline to half of the initial value.

**Table 4 ijerph-19-01365-t004:** Acute toxicity test towards *V. fischeri* before and after 100 d of incubation of R and PLD soils contaminated with diuron and treated with *O. anthropi* CD3 and HPBCD.

Soil	Treatment	EC_50_ (%) *	TU	Class	Toxicity **
R	Uncontaminated	166	0.6	Class II	Slight acute toxicity
	Contaminated with diuron	24.7	4.0	Class III	Acute toxicity
	*O. anthropi* CD3	30.8	3.2	Class III	Acute toxicity
	*O. anthropi* CD3 + HPBCD	29.9	3.3	Class III	Acute toxicity
PLD	Uncontaminated	111	0.9	Class II	Slight acute toxicity
	Contaminated with diuron	9.2	10.9	Class IV	High Acute toxicity
	*O. anthropi* CD3	73.2	1.4	Class III	Acute toxicity
	*O. anthropi* CD3 + HPBCD	72.8	1.4	Class III	Acute toxicity

* For each soil sample the EC_50_ value corresponds to the soil extract concentration (% *v*/*v*) having a toxic effect on 50% of *V. fischeri*. ** According to Persoone et al. (2003).

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
