# Peer review of "Enhanced Biodegradation of Phenylurea Herbicides by Ochrobactrum anthrophi CD3 Assessment of Its Feasibility in Diuron-Contaminated Soils"

_ijerph, 2022, doi:10.3390/ijerph19031365_

Round 1

Reviewer 1 Report

Comments and suggestions for Authors

”Enhanced biodegradation of phenylurea herbicides by Ochrobactrum antrophi (?) CD3. Assessment of its feasibility in diuron contaminated soils”

The subject is interesting and fall within the scope of the journal. The experimental dataset undoubtedly are useful  and constitutes scientific values. The presented manuscript deals with the current global problem.

In the present study, CD3 strain was tested for diuron, linuron, chlorotoluron, isoproturon and fluometuron biodegradation in solution. Furthermore, the investigated strain was inoculated in two soil artificially contaminated  with diuron, with the aim of finding an effective bioremediation tool based on biostimulation (using nutrients solution (NS)), bioaugmentation (inoculating with O. anthropi CD3) and the use of 2-hydroxypropyl-β-cyclodextrin (HPBCD) which is able to improve water solubility of the herbicide diuron and hence, its bioavailability to be degraded. An ecotoxicology study was also carried out before and after soil diuron bioremediation inoculating O. anthropi CD3 to determine if the treatment achieved a decreased the soil toxicity.

General remarks

The following points may be addressed by the Authors to enhance the usefulness of the paper.

  • The applied research methods are correct and do not raise my objections.
  • The obtained results were described correctly. Literature references have been used correctly.
  • Lines 294-295 >There was no evidence for their abiotic dissipation in the uninoculated controls (data not shown)< . This statement should be deleted as no supporting data was provided.
  • Figures 3 and 5. Descriptions of subsequent drawings should be completed, and the soils should be marked.
  • The conclusions were presented correctly and reliably.

The research should be continued in natural microfield conditions, on soils of various categories.

Specific comments

Line 2 – should be anthropi

Line 69 – is [22] – should be [21]?

Line 130 – is 2.1….. – should be 2.2….

Line 150 – is 2.1….. – should be 2.3…..

Line 157 – is 2.1….. – should be 2.4…..

Line 163 – is 2.1….. – should be 2.5…..

Line 172 – is 2.1….. – should be 2.6…..

Line 184 – is 2.1….. – should be 2.7…..

Line 205 – is 2.1….. – should be 2.8…..

Line 222 – is 2.1….. – should be 2.9…..

Line 230 – is 2.1….. – should be 2.10…..

Line 243 – is 2.1….. – should be 2.11…..

Line 263 – is 2.….. – should be 3…..

Line 403 – should be Table 3. Kinetic……

The font and size should be standardized throughout the manuscript.

The References  must be adapted to the publishing requirements.

Author Response

ANSWER TO REVIEWER

Reviewer 1:
General remarks
- The following points may be addressed by the Authors to enhance the usefulness of the paper.
- The applied research methods are correct and do not raise my objections. 
- The obtained results were described correctly. Literature references have been used correctly. 
Point 1. Lines 294-295 >There was no evidence for their abiotic dissipation in the uninoculated controls (data not shown)< . This statement should be deleted as no supporting data was provided.  
Response 1: Lines 294-295 have been removed.
Point 2. Figures 3 and 5. Descriptions of subsequent drawings should be completed, and the soils should be marked.
Response 2: Changes have been done in the figures.
- The conclusions were presented correctly and reliably.
- The research should be continued in natural microfield conditions, on soils of various categories.
Specific comments
Point 3. Line 2 – should be anthropi. 
Response 3: The correction has been made.
Point 4. Line 69 – is [22] – should be [21]? 
Response 4: The correction has been made.
Point 5. Line 130 – is 2.1….. – should be 2.2…. 
Response 5: The correction has been made.
Point 6. Line 150 – is 2.1….. – should be 2.3….. 
Response 6: The correction has been made.
Point 7. Line 157 – is 2.1….. – should be 2.4….. 
Response 7: The correction has been made.
Point 8. Line 163 – is 2.1….. – should be 2.5….. 
Response 8: The correction has been made.
Point 9. Line 172 – is 2.1….. – should be 2.6….. 
Response 9: The correction has been made.
Point 10. Line 184 – is 2.1….. – should be 2.7….. 
Response 10: The correction has been made.
Point 11. Line 205 – is 2.1….. – should be 2.8….. 
Response 11: The correction has been made.
Point 12. Line 222 – is 2.1….. – should be 2.9….. 
Response 12: The correction has been made.
Point 13. Line 230 – is 2.1….. – should be 2.10….. 
Response 13: The correction has been made.
Point 14. Line 243 – is 2.1….. – should be 2.11….. 
Response 14: The correction has been made.
Point 15. Line 263 – is 2.….. – should be 3….. 
Response 15: The correction has been made.
Point 16. Line 403 – should be Table 3. Kinetic…… 
Response 16: Modifications has been conducted.
Point 17. The font and size should be standardized throughout the manuscript. 
Response 17: Font and size have been standardized.
Point 18. The References must be adapted to the publishing requirements. 
Response 18: References have been adapted to the publishing requirements.

Reviewer 2 Report

It is a very good document, very well written, and generally well presented. Their results are interesting and with great application potential. I consider that the suggested adjustments should be published.

I have the following suggestions to improve the manuscript:

Introduction

This chapter correctly covers the essential aspects, it is very well approached.

Line 32: necrosis and plant death (5)

Line 46: to include one reference

Line 62: to include one reference

Line 71: …soils (22), and …

Introduction has 40 references, that's a lot for a research article, is it possible to eliminate some?

Materials and methods

Methods are very well described.

Table 1. in the PDL soil the sum of % of Sand, silt and clay gave 98.6% should give 100%, check

Line 137: subscript in H2O

Missing to include the experimental design, for example completely randomized design

Results and discussion

The results are well described, I consider that the discussion in some cases is very broad and they do not always relate it to their results

Line 625-626: delete this sentence

Line 325-330. This part of the discussion needs to be much more concrete

Line 331: What implication does this result have from a practical point of view?

Line 334-343: the results reported there from the study of reference 21 must be related to its results, otherwise it is only state of the art. I suggest to summarize this paragraph.

Line 344-354: same suggestion above

Line 490: include more discussion. How is the increase in 3.4-DCA in the first hours in Figure 5, and after 24d in Figure 5A, explained?

Lines 492-502: this information should go after mentioning the results. In addition, this information should be summarized

Figures:

I consider that the presentation of the figures should be improved so that it is consistent with the high quality presented by the rest of the manuscript.

Figure 1: include tick marks on the axes. Include in the title of the figure that they are the vertical bars, SE?

Figure 2. The same suggestions as figure 1

Figure 3. In addition to the suggestions above, the legend in the figure above is incomplete. Include figure 3A and Figure 3B, describe in the title of the figure

Figure 4. Remove the legends and include the treatments on the "x" axis, differentiate Figure 4A and Figure 4B. Missing title of a “y” axis. Can a comparison of means test be included to determine differences between treatments?

Fig. S1. Remove the legends and include the treatments on the "x" axis. Can a comparison of means test be included to determine differences between treatments? include tick marks on the axes. Include in the title of the figure that they are the vertical bars, SE?

Author Response

ANSWER TO REVIEWER

Reviewer 2:
Introduction
- This chapter correctly covers the essential aspects, it is very well approached.
Point 1: Line 32: necrosis and plant death (5) 
Response 1: The correction has been made (Line 32).
Point 2: Line 46: to include one reference. 
Response 2: Reference has been included (Line 46).
Point 3: Line 62: to include one reference. 
Response 3: Reference has been included (Line 62).
Point 4: Line 71: …soils (22), and … 
Response 4:  Line 72 Sentence was modified.
Point 5: Introduction has 40 references, that's a lot for a research article, is it possible to eliminate some? 
Response 5: Several references have been removed from the manuscript.
Materials and methods
- Methods are very well described.
Point 6: Table 1. in the PDL soil the sum of % of Sand, silt and clay gave 98.6% should give 100%, check. 
Response 6: Data were revised.
Point 7: Line 137: subscript in H2O. 
Response 7: The correction has been made (Line 142).
Point 8: Missing to include the experimental design, for example completely randomized. 
Response 8: Information has been added in Line 192-193.
Results and discussion
Point 9: The results are well described, I consider that the discussion in some cases is very broad and they do not always relate it to their results
Response 9: In some cases, the discussion has been summarized and non-relevant information has been removed from the manuscript.
Point 10: Line 625-626: delete this sentence. 
Response 10: These lines correspond to reference. 
Point 11: Line 325-330. This part of the discussion needs to be much more concrete. 
Response 11: The correction has been made (line 340).
Point 12: Line 331: What implication does this result have from a practical point of view?
Response 12: Answer has been included in lines 349-351 
Point 13: Line 334-343: the results reported there from the study of reference 21 must be related to its results, otherwise it is only state of the art. I suggest to summarize this paragraph.
 Response 13: Paragraph has been summarized (Lines 352-364).
Point 14: Line 344-354: same suggestion above. 
Response 14: Paragraph has been summarized (Lines 365-370).
Point 15: Line 490: include more discussion. How is the increase in 3.4-DCA in the first hours in Figure 5, and after 24d in Figure 5A, explained?
Response 15: Explanation is included in Lines 498 – 4500. OM content is higher in R soil than PLD soil, and hence, the diuron bioavailability is lower provoking a slower biodegradation of the herbicide. 
Point 16: Lines 492-502: this information should go after mentioning the results. In addition, this information should be summarized. 
Response 16: The information has been moved and summarized (Lines 551-555).
Figures:
Point 17: I consider that the presentation of the figures should be improved so that it is consistent with the high quality presented by the rest of the manuscript.
Response 17: Figures have been improved based on your suggestions.
Point 18: Figure 1: include tick marks on the axes. Include in the title of the figure that they are the vertical bars, SE?
Response 18: Figure 1 has been modified.
Point 19: Figure 2. The same suggestions as figure 1
Response 19: Figure 2 has been modified.
Point 20: Figure 3. In addition to the suggestions above, the legend in the figure above is incomplete. Include figure 3A and Figure 3B, describe in the title of the figure
Response 20: Figure 3 has been modified.
Point 21: Figure 4. Remove the legends and include the treatments on the "x" axis, differentiate Figure 4A and Figure 4B. Missing title of a “y” axis. Can a comparison of means test be included to determine differences between treatments?
Response 21: Figure 4 has been modified in based on suggestions.
Statistical analysis has been conducted using analysis of variance (ANOVA). In addition, Statistical analysis has been included in materials and methods section (lines 271-274)
Point 22: Fig. S1. Remove the legends and include the treatments on the "x" axis. Can a comparison of means test be included to determine differences between treatments? include tick marks on the axes. Include in the title of the figure that they are the vertical bars, SE?
Response 22: Figure S1 has been modified in based on suggestions.
Statistical analysis has been conduct using analysis of variance ANOVA.
